# Long wavelength near-infrared and red light-driven consecutive photo-induced electron transfer for highly effective photoredox catalysis

Le Zeng [1,2,3,4], Ling Huang[1,3], Zhi Huang[3], Tomoyasu Mani [5], Kai Huang[1], Chunying Duan [2] ✉ & Gang Han [1] ✉

Consecutive photoinduced electron transfer (conPET) processes accumulate the energies of two photons to overcome the thermodynamic limit of traditional photoredox catalysis. However, the excitation wavelength of conPET systems mainly focused on short wavelength visible light, leading to photodamage and incompatibility with large-scale reactions. Herein, we report on conPET systems triggered by near-infrared (NIR) and red light. Specifically, a blue-absorbing conPET photocatalyst, perylene diimide (PDI) is sensitized by a palladium-based photosensitizer to triplet excited state ($^3$PDI$^*$), which generates PDI radical anion (PDI$^{\cdot-}$) over 100-fold faster than that in the conventional conPET. Accordingly, photoreduction with superior reaction rate and penetration depth, as well as reduced photodamage is detected. More importantly, our work offers comprehensive design rules for the triplet-mediated conPET strategy, whose versatility is confirmed by metal-free dye pairs and NIR-active PtTNP/PDI. Notably, our work achieves NIR-driven atom transfer radical polymerization using an inert aromatic halide as the initiator.

Photosynthesis is the primary process by which solar energy is converted into chemical energy in nature[1–3]. Drawing inspiration from this natural phenomenon, chemists have developed the field of photocatalysis[4,5], which uses light energy to create chemical bonds. However, there are some key differences between natural photosynthesis and chemical photocatalysis. In photosynthesis, consecutive photon absorption occurs, while chemical photocatalysis typically rely on single-photon excitation to drive reactions[6,7]. This difference limits the reaction scope of traditional visible light-induced photoredox catalysis, as it can only activate substrates with mild activation energy[8]. Consequently, many synthetically important and thermodynamically demanding transformations, particularly those involving the

activation of inert bonds, require ultraviolet excitation[9–11]. This approach can lead to photodamage and undesired side reactions, presenting challenges for efficient and selective photocatalytic processes.

Recently, the multi-photon excitation strategy has gained significant interest as a way to overcome the thermodynamic limitations of traditional visible photocatalysis[11,12]. This approach involves accumulating the energies of two or more photons for one catalytic cycle. A crucial aspect of multi-photon excitation is the long-lived intermediate that can temporarily store energy. The consecutive photoinduced electron transfer (conPET) process has emerged as one of the most extensively investigated and feasible multi-photon excitation

[1]Department of Biochemistry and Molecular Biotechnology, University of Massachusetts Chan Medical School, Worcester, MA, USA. [2]State Key Laboratory of Fine Chemicals, Dalian University of Technology, Dalian, PR China. [3]Tianjin Key Laboratory of Biosensing and Molecular Recognition, Research Center for Analytical Sciences, College of Chemistry, Nankai University, Tianjin, PR China. [4]School of Materials Science and Engineering, Nankai University, Tianjin, PR China. [5]Department of Chemistry, University of Connecticut, Storrs, CT, USA. ✉e-mail: cyduan@dlut.edu.cn; Gang.Han@umassmed.edu

strategies for photoredox catalysis. This is due to the relatively long lifetime of its energy-storing intermediate, the radical anion of the photocatalyst (PC$^{\bullet-}$). In conPET, two separate photoinduced electron transfer (PET) steps are required. In the first PET step, the photocatalyst (PC) absorbs a photon and is excited to its singlet excited state ($^1$[PC]$^*$), which can gain an electron from an electron donor to produce the energy-storing intermediate PC$^{\bullet-}$ (Fig. 1a). In the second PET step, the ground state PC$^{\bullet-}$ absorbs another photon, reaching its excited state ($^2$[PC$^{\bullet-}$]$^*$). This excited state transfers its high-energy electron to a Supplementary Table substrate, completing the catalytic cycle. By partially mimicking biological photosynthesis, the conPET process acts as an energy ladder building up energy to enable transformations beyond the reach of traditional single-photon photocatalysis[13–15]. One example of this is the visible light-mediated photoreduction of inactivated aryl chlorides[13].

Despite recent advances, existing conPET processes still face limitations due to their reliance on the singlet excited state of photocatalysts like perylene diimide (PDI)[13], Rhodamine 6G[16], acridinium salt[15], and benzo[ghi]perylene imides[17]. As a result, current conPET systems are restricted to the inherent absorption of these photocatalysts. The major excitation source for conPET catalysis has been blue light and reserachers also suggest that the absorption tail of the conPET photocatalyst may shift the wavelength to the green region[13,18]. These fundamental limitations lead to several significant drawbacks, such as suboptimal energy-utilization efficiency, incompatibility with large-scale reactions, and the potential for photobleaching and photodamage to reactants[19,20].

Recent efforts on long-wavelength light-driven photocatalytic reactions suggested the potential to overcome these limitations with two-photon involved processes. For example, photosensitizers

### a  Reported conPET process for photoreduction of aryl halides

### b  This work: Triplet sensitization mediated conPET process for photoreduction of aryl halides

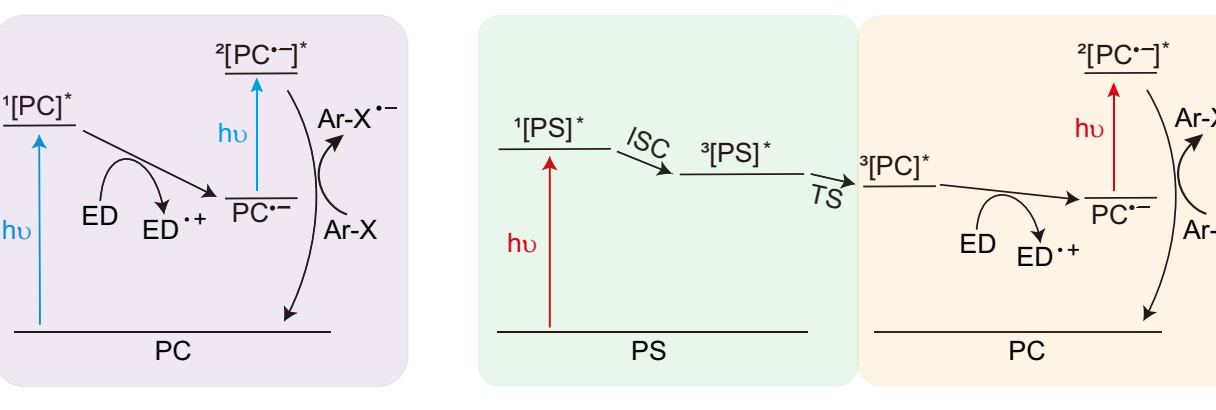

### c

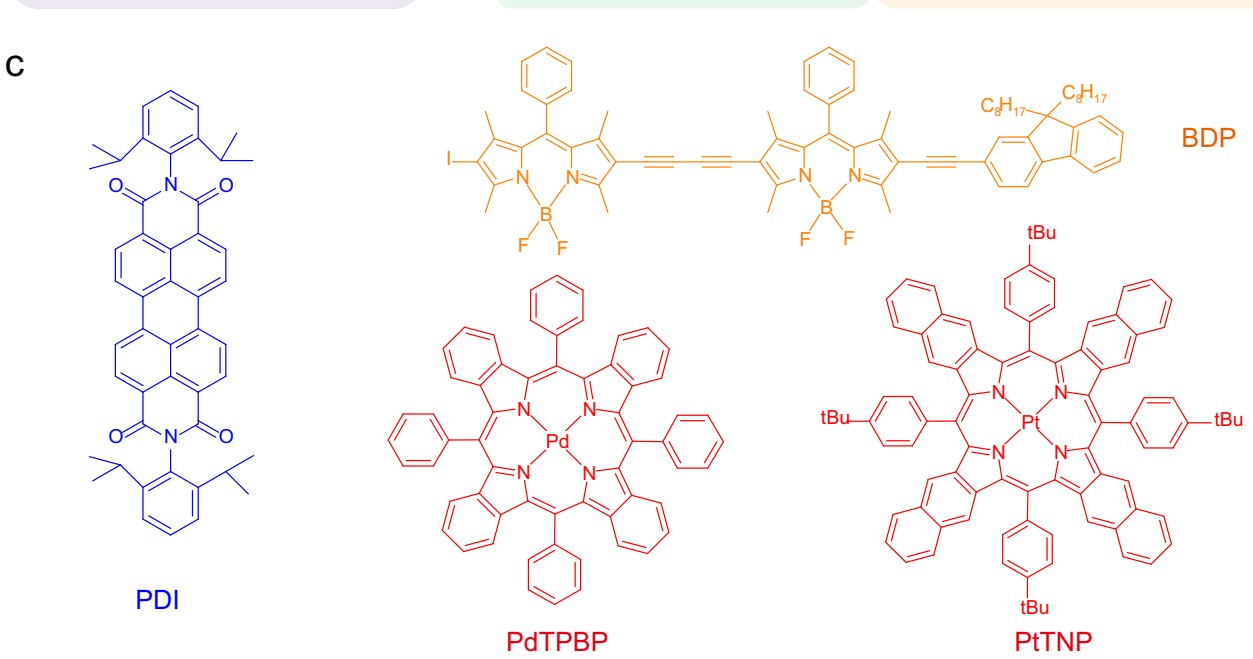

**Fig. 1 | The proposed triplet sensitization-mediated consecutive photoinduced electron transfer (TS-conPET) process for red or NIR light-driven photoreduction of aryl halides.** a The reported consecutive photoinduced electron transfer (conPET) process for the photoreduction of aryl halides. PC photocatalyst, ED electron donor, Ar-X aryl halide. $^1$[PC]$^*$, the excited singlet state of PC; PC$^{\bullet-}$, the radical anion of PC; $^2$[PC$^{\bullet-}$]$^*$, the excited doublet state of PC$^{\bullet-}$; ED$^{\bullet+}$, the radical cation of ED; Ar-X, aryl halide; Ar-X$^{\bullet-}$, the radical anion of Ar-X. hυ, light irradiation. b Long wavelength light-driven TS-conPET process for the photoreduction of aryl halides. PS photosensitizer, ISC intersystem crossing, TS triplet sensitization. $^3$[PS]$^*$, the excited triplet state of PS. c Molecular structures of the photocatalyst PDI and the photosensitizers PdTPBP, BDP, and PtTNP.

possessing two-photon absorption (TPA) capability such as ruthenium polypyridyl complexes[21] and benzothiazole derivatives[22] were reported to accomplish $^1O_2$-mediated energy transfer reactions and photoredox reactions including hydrodehalogenation, C−H cyanation, and Ni-catalyzed allylation of aldehydes upon NIR irradiation (740 nm or 850 nm). Besides, triplet fusion upconversion with sensitizer/annihilator pairs of organic dyes[19] or lead-free nanocrystals[23] were utilized to trigger high-energy transformations (hydrodehalogenation, reductive radical cyclization, intramolecular [2 + 2] cyclization, vinyl azide reduction, radical and cationic polymerization) with NIR irradiation. These NIR-driven photocatalytic systems displayed comparable and even higher reaction yields, better reaction selectivity, and broader substrate scope than their shorter-wavelength counterpart, due to the deeper penetration depth and less competing absorption of substrate or cocatalyst for NIR.

Alternatively, for conPET process, there are implications for the exploitation of light source beyond the intrinsic absorption of conPET moiety. Goez M. et al. found that green-light excited [Ru(bpy)$_3$]$^{2+}$ can sensitize pyrene-1-carboxylate (Py$^-$) to its excited state, which can generate the radical anion Py$^{2-}$ for subsequent excitation to produce highly-reducing hydrated electrons for chloroacetate decomposition[24]. However, this interesting process is incompatible with organic solvents due to the instability of Py$^{2-}$, restricting its potential for conPET-related photoredox catalysis[25]. Besides, during the preparation of our manuscript, Wenger et al. reported that the combination of a red light-absorbing Cu complex and a classic conPET photocatalyst, 9,10-dicyanoanthracene (DCA) is efficient to drive dehalogenation and detosylation reactions upon 623 nm irradiation[26]. Nevertheless, the complicate mechanistic pathways of this system make it difficult to expand to other combinations. These results indicated that long-wavelength light-driven conPET systems for photocatalytic applications is feasible and there is urgent demand for its development and rational design.

In response to these limitations, we propose a triplet sensitization-mediated conPET (TS-conPET) mechanism to capture and relay long-wavelength photon energy. As shown in Fig. 1b, the TS-conPET system comprises two components: a photosensitizer (PS) and a photocatalyst (PC). First, the PS absorbs a long-wavelength photon to generate a singlet excited PS ($^1$[PS]$^*$), which undergoes intersystem crossing to obtain a triplet excited PS ($^3$[PS]$^*$). This energy is then transferred to the triplet excited PC ($^3$[PC]$^*$) via triplet sensitization. Unlike the conventional conPET, which starts with the singlet excited state of the PC, in TS-conPET system the generated $^3$[PC]$^*$ would receive an electron from the reducing agent to produce PC$^{\cdot-}$ to complete the first PET step. The generated PC$^{\cdot-}$ can subsequently absorb the second photon of long-wavelength light to reach [PC$^{\cdot-}$]$^*$ and then conduct the second PET with photocatalytic substrates. By utilizing this TS-conPET mechanism, we aim to overcome the drawbacks of existing conPET processes and enable the development of more versatile and efficient photocatalytic systems capable of harnessing a wider range of light wavelengths.

Herein, we validate the operation of TS-conPET with the well-known conPET photocatalyst PDI and various red or NIR photosensitizers including PdTPBP, BDP, and PtTNP (Fig. 1c). Transient absorption spectra, phosphorescence quenching spectra and absorption spectra analysis clearly show the generation of the triplet excited state of PDI ($^3$PDI$^*$) and PDI radical anion (PDI$^{\cdot-}$) upon red (625/650 nm) or NIR (721 nm) light irradiation, with the assistance of photosensitizer. Excitingly, compared to that in conventional conPET under blue light irradiation, the generation rate of PDI$^{\cdot-}$ under long wavelength light excitation with our triplet sensitization method is over two orders of magnitude (102 times) faster, which might benefit from the longer lifetime of triplet excited state and the larger cage escape yield of triplet geminate radical pair in our method, compared to the singlet excited state and the singlet geminate radical pair in the traditional

conPET process[27]. Such triplet sensitization-generated PDI$^{\cdot-}$ is found to be able to realize long-wavelength light-driven photoreduction of aryl halides, suggesting the operation of TS-conPET, which exhibiting faster reaction rate than that of conventional blue light-mediated conPET, especially for large-volume reaction (20 mL). Furthermore, the aromatic radical generated via this TS-conPET approach is utilized for atom transfer radical polymerization (ATRP) with metal-free dye pairs or NIR-active dye pairs.

## Results and discussion

### Triplet sensitization of PDI by PdTPBP

Initially, the widely-investigated conPET photocatalyst PDI[13,28] and red-light photosensitizer PdTPBP[29,30] were paired to explore the feasibility of the TS-conPET cycle. The photophysical properties of PdTPBP and PDI were firstly examined and recorded (Supplementary Fig. 1 and Supplementary Table 1). PdTPBP showed a sharp Q band absorption peak at 628 nm with a molar extinction coefficient ($\varepsilon$) of $1.1 \times 10^5 M^{-1}$ cm$^{-1}$ (Supplementary Fig. 1)[29]. In contrast, PDI presented symmetrical absorption and emission spectra, with peaks at 527 and 545 nm, respectively. As the absorption peak of PDI tails off beyond 550 nm (Supplementary Fig. 1), wavelengths shorter than 550 nm are necessary to activate PDI to its singlet excited state (S$_1$). The phosphorescence emission of PdTPBP was detected at 790 nm (Supplementary Fig. 1), implying that the triplet excited state (T$_1$) energy of PdTPBP can be calculated to be 1.57 eV. On the other hand, the T$_1$ energy of PDI was reported to be 1.21 eV[30], which is lower than that of PdTPBP. Hence, we anticipate an energetically favorable exothermic triplet-triplet energy transfer (TTET) between PdTPBP and PDI.

To experimentally confirm the TTET between PdTPBP and PDI, we studied the nanosecond transient absorption (TA) spectra of PdTPBP with and without PDI in DMF. Upon pulsed laser excitation at 355 nm (4 mJ), we detected the ground-state bleaching (GSB) bands of PdTPBP at 420–450 nm and 600–650 nm, which are consistent with the steady-state absorption of PdTPBP (Supplementary Figs. 1 and 2a). The excited-state absorption (ESA) bands of PdTPBP, which can be attributed to the transitions from T$_1$ to T$_n$[29], were observed at 460–570 nm. In the presence of PDI, the ESA shoulder peaks of PdTPBP gradually turned to a broad featureless band, which start to show the characteristic ESA peaks of $^3$PDI$^*$ (470 and 490 nm)[30–32] after delay time of 20 μs (Fig. 2a, b, $\lambda_{ex}$ = 630 nm). At the same time, the GSB bands of PDI were detected for the PdTPBP/PDI mixture upon irradiating with either 355 or 630 nm laser (Fig. 2 and Supplementary Fig. 2), which further verify the triplet sensitization of PDI by the triplet excited state of PdTPBP ($^3$PdTPBP$^*$). We also observed the evolution of $^3$PDI$^*$ via monitoring the decay traces of PdTPBP's ESA signal (535 nm, Fig. 2c) and GSB signal (640 nm, Supplementary Fig. 3). The lifetime of $^3$PdTPBP$^*$ was found to be shortened in the presence of PDI, and increasing the PDI concentration shortened it more severely, indicating more efficient TTET with higher PDI concentration (Fig. 2c and Supplementary Table 2). Additionally, the decay trace of $^3$PdTPBP$^*$ at 535 nm showed a reverse curve upon PDI addition, occurring at an earlier time-point with an increased PDI ratio (160 μs for PdTPBP/PDI 1:1 and 110 μs for PdTPBP/PDI 1:3). The long tail of the reverse curve indicated that a species with a long lifetime had been generated, which was also observed with the GSB decay traces of PdTPBP at 640 nm (Supplementary Fig. 3 and Supplementary Table 3). These results confirm that the energy of $^3$PdTPBP$^*$ can be transferred to PDI to induce the generation of $^3$PDI$^*$.

To further study the TTET process of PdTPBP/PDI pair, the phosphorescence change of PdTPBP upon the addition of PDI was used to calculate the Stern-Volmer constant (k$_{sv}$)[28]. As seen in Fig. 2d, notable phosphorescence quenching of PdTPBP occurred upon the consecutive addition of PDI. The calculated k$_{sv}$ ($7.2 \times 10^5 M^{-1}$) and bimolecular quenching constant (k$_q$, $4.88 \times 10^9 M^{-1} s^{-1}$) of PdTPBP/PDI illustrated that the triplet excited energy of PdTPBP can efficiently

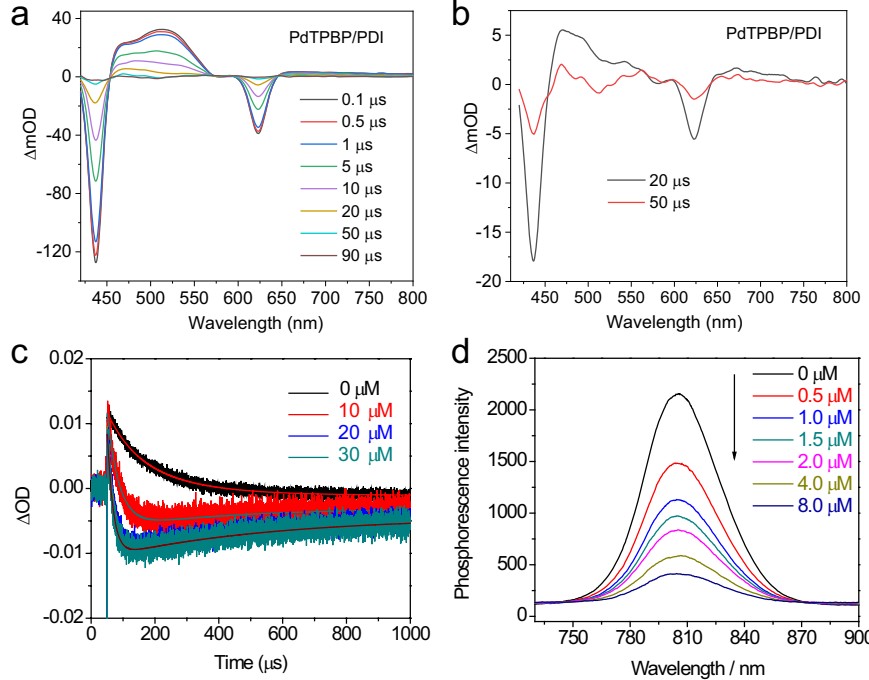

**Fig. 2 | The evidence of the triplet sensitization of PDI by PdTPBP. a** Nanosecond transient absorption spectra of PdTPBP/PDI in DMF at different delay times (0.1–90 μs) and the selected two traces (**b**). $c$ (PdTPBP) = $c$ (PDI) = 10 μM, $\lambda_{ex}$ = 630 nm. **c** Decay traces at 535 nm of PdTPBP/PDI with different PDI concentration (0−30 μM). $c$ (PdTPBP) = 10 μM, $\lambda_{ex}$ = 355 nm. **d** Phosphorescence quenching of PdTPBP via titration of PDI. $c$ (PdTPBP) = 10 μM, in deaerated DMF, $\lambda_{ex}$ = 630 nm.

transfer to PDI in DMF. Meanwhile, the quantum efficiency of TTET ($\Phi_{TTET}$) is determined to be 76.0% from PdTPBP (10 μM) to PDI (8 μM) in DMF. Due to the electron-deficient nature of PDI, the TTET process from PdTPBP to PDI may be influenced by solvent polarity and the presence of electron donor[33,34]. Fortunately, in our study, $\Phi_{TTET}$ from PdTPBP to PDI in the polar solvent of DMF (76.0%) is just slightly lower than that in the nonpolar solvent of toluene (84.3%) (Supplementary Fig. 4 and Supplementary Table 4), indicating that the solvent polarity has an insignificant effect on the TTET of PdTPBP/PDI. In the presence of electron donor triethylamine (TEA, 1 mM) or the substrate 4-bromoacetophenone (1 mM), the $k_{sv}$, $k_q$ and $\Phi_{TTET}$ values of PdTPBP/PDI remain at nearly the same level (Supplementary Table 4 and Supplementary Figs. 4–7). These experimental results suggested that PdTPBP/PDI can conduct an efficient TTET process under different conditions to produce $^3$PDI* (Fig. 2). Since molecular self-assembly is known to facilitate intermolecular triplet energy transfer[35], we also studied the micro-structure of the PdTPBP/PDI pair via a transmission electron microscope (TEM). Interestingly, we observed the formation of spherical nanoparticles for PdTPBP/PDI (Supplementary Fig. 8). As a control, we employed leucine-modified PDI (Leu-PDI) as a photocatalyst (Supplementary Fig. 9), which didn't form nanoparticles in the presence of PdTPBP (Supplementary Fig. 10), mainly due to the charge repulsion effect of the carboxylic acid groups from Leu-PDI. This new pair PdTPBP/Leu-PDI exhibited a smaller $\Phi_{TTET}$ than that of PdTPBP/PDI (53.7% vs 76.0 %) (Supplementary Fig. 11). These results indicated that self-assembly is likely due to the π-π stacking of PdTPBP and PDI or between PDI molecules, which contributes to the mutual interaction between PdTPBP and PDI toward more efficient triplet sensitization.

### Generation of PDI radical anion from the triplet state of PDI
After discovering the highly effective TTET between PdTPBP and PDI, we then explored whether the generated $^3$PDI* can receive an electron from TEA to produce the radical anion (PDI$^{·-}$), which is the key intermediate of the conPET process. The mixed solution of PDI, PdTPBP and

TEA were kept in an oxygen-free environment and its absorption spectra were monitored upon red-light irradiation (625 nm LED). After only 10 s of irradiation, clear PDI$^{·-}$ fingerprint peaks[13,28] ranging from 600 nm to 900 nm appeared in the spectra (Fig. 3a), and these peaks continued to grow with prolonged irradiation time. In the meantime, the original yellow-brown solution turned dark-green, which also suggests the generation of PDI$^{·-}$. As expected, when slowly introducing air into this dark-green solution, the absorption of PDI$^{·-}$ gradually weakens and eventually disappears (Fig. 3b). The color of the solution gradually returned to its original yellow-brown. For the control experiment in the absence of PdTPBP, only negligible PDI$^{·-}$ absorption can be detected (Fig. 3c), which may result from the background ambient light irradiation. Besides, for the ns-TA spectra, when the donor molecule Et$_3$N was added, a newly-emerged GSB peak at 700 nm was detected for PdTPBP/PDI mixture (Supplementary Fig. 13), suggesting the generation of PDI$^{·-}$. These results demonstrated that PDI$^{·-}$ can be obtained in the presence of TEA after triplet sensitization under 625 nm irradiation.

In previous conPET studies, PDI was excited by blue light (455 nm) to reach $^1$PDI*, which then received one electron from TEA to generate PDI$^{·-}$[13]. By contrast, we verified that in our system PDI$^{·-}$ was generated from $^3$PDI* rather than $^1$PDI* via precluding the possibility of the PdTPBP/PDI pair to proceed triplet-triplet annihilation upconversion (TTA-UC) for the generation of $^1$PDI*. This result is clearly distinct from the existing conPET mechanism. In particular, the TTA-UC emission of PdTPBP/PDI was detected to be extremely sensitive to the polarity of solvents. In toluene, intense TTA-UC emission centered at 545 and 581 nm with a TTA-UC efficiency of 3.4% was observed for PdTPBP/PDI, when excited by 650 nm light (Supplementary Fig. 14). However, with increased solvent polarity, the TTA-UC emission intensity of PdTPBP/PDI decreased significantly. More specifically, in the DMF solvent that we used for the triplet sensitization study, the TTA-UC emission of PdTPBP/PDI was completely quenched. These results verified that, for the PdTPBP/PDI pair in DMF, PDI$^{·-}$ was generated from $^3$PDI* via a triplet

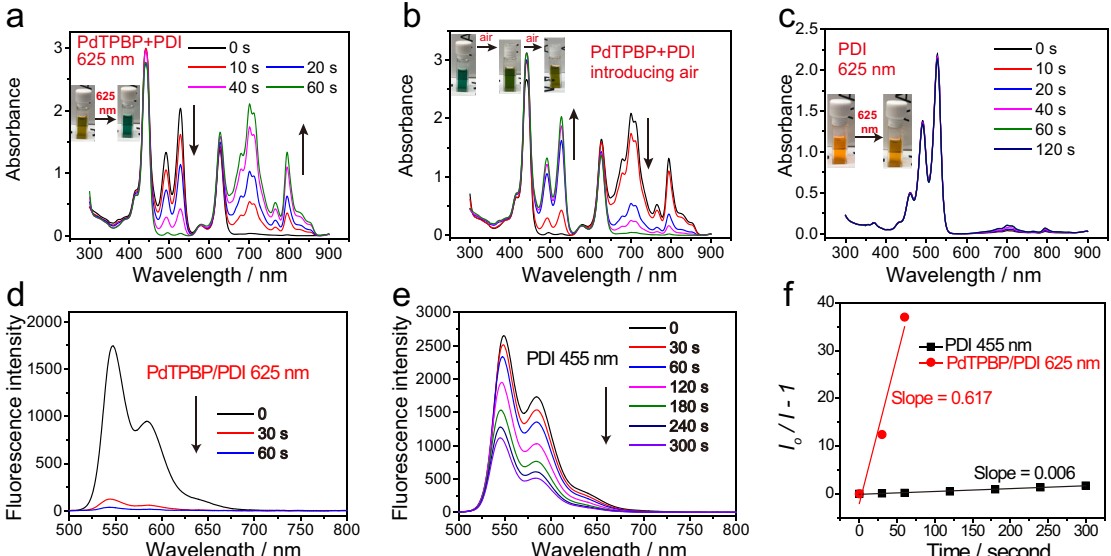

**Fig. 3 | Efficient generation of PDI⁻ via red light-driven triplet sensitization.** UV-vis absorption changes of solutions in the presence of electron donor Et₃N: **a** the mixture of PDI (50 μM) and PdTPBP (10 μM) upon red illumination in Ar; **b** introducing air into treated (**a**); **c** PDI (50 μM) upon red illumination in Ar. For (**a**–**c**), the insets are the solution images upon specific treatment. **d** Fluorescence emission spectra of PdTPBP/PDI in the presence of Et₃N under red illumination. **e** Fluorescence emission spectra of PDI in the presence of Et₃N under blue illumination. **f** The fitting curves of PDI⁻ generation rate under red (PdTPBP/PDI) or blue illumination (PDI) and the respective slope are given. $I_0$ and $I$ represent the original absorption and the real-time absorption at specific timepoint, respectively. For (**a**–**f**), the solvents are all DMF. Red illumination refers to 625 nm LED (20 mW/cm²), while blue illumination refers to 455 nm LED (20 mW/cm²).

sensitization approach rather than from ¹PDI* via an indirect TTA-UC emission re-absorption process.

After confirming the successful generation of PDI⁻, the generation rate of PDI⁻ via this triplet sensitization strategy (625 nm illumination, PdTPBP/PDI) was calculated by measuring the change of PDI fluorescence intensity. This can be done because PDI is highly fluorescent meanwhile PDI⁻ is non-fluorescent[36]. Similarly, the generation rate of PDI⁻ via blue light mediated conPET was also calculated. For the PdTPBP/PDI pair (optical density, OD 0.9, 625 nm) with red light illumination (625 nm, 20 mW/cm²), 97.4% fluorescence intensity of PDI was quenched within 60 s in the presence of TEA in DMF (Fig. 3d). On the contrary, under blue light illumination (455 nm, 20 mW/cm²), the fluorescence intensity of PDI (OD 0.9 at 455 nm) quenched much more slowly, only 63.3% within 300 s (Fig. 3e). In addition, we found that the fluorescence of PDI with TEA was not affected by 625 nm light illumination (Supplementary Fig. 15), confirming that PDI⁻ generation under 625 nm light irradiation requires the presence of PdTPBP. We then fitted the fluorescence quenching rates and found that PdTPBP-sensitized PDI⁻ generation under red light illumination was remarkably quicker (102-fold) than that of conventional conPET under blue light illumination. Moreover, the generation rates of PDI⁻ under different excitation powers (40, 60, and 80 mW/cm²) via our red light driven TS-conPET approach or blue light-driven conventional conPET approach were calculated and compared. As shown in Supplementary Fig. 16, under all these power densities, our TS-conPET approach was faster than conventional conPET in regard to the production of PDI⁻.

**Red light-driven photoreduction via triplet sensitization-mediated conPET and its superiority over traditional conPET**

The photoactivation of aryl halides is a crucial topic in organic synthesis because the generated aromatic radical is a building block for various molecules of synthetic and pharmaceutic importance[10,11]. However, traditional single-photon processes using UV light are typically required to activate inert aryl halides due to their negative reduction potential ($E_{red} < -1.8$ V)[8,10]. The conPET process has been explored to shift the excitation wavelength for inert aryl halides activation from UV to visible light[13], but as shown in Supplementary

Table 6, the reported excitation wavelength for a traditional conPET process is limited to one shorter than 530 nm. Alternatively, the TTA-UC process has been attempted for the photoreduction of inert aryl halides via the accumulation of the energies of two photons[37–39]. Several TTA-UC pairs have been reported to conduct the photoreduction of aryl bromides. Chemists have also developed other methods to overcome the energy shortage for the photoreduction of inert aryl halides, such as generating the radical anion of the photocatalyst through electrocatalysis and then exciting it with blue LEDs to trigger the photoreduction of aryl chlorides[40]. In addition, triplet sensitization has been utilized for the photoreduction of aryl halides[25]. For instance, the photocatalyst pyrene, which can only absorb light with a wavelength shorter than 350 nm, was sensitized to its excited state by Ru(bpy)₂³⁺ upon blue light illumination and then produced the radical anion to trigger the photoreduction of aryl halides[25]. However, since the radical anion of pyrene is unstable, this system cannot absorb another photon to undergo the conPET process. Overall, the long-wavelength light-driven photoreduction of inert aryl halides remains a key challenging area of research.

In this regard, after confirming the superior PDI⁻ generation rate via the triplet sensitization of PdTPBP/PDI pair with 625 nm LED, we studied whether the generated PDI⁻ can absorb another 625 nm photon to complete the TS-conPET cycle for the photoreduction of inert aryl halides (Fig. 1b). We chose 4-bromoacetophenone as the model substrate. A product yield of 47% was recorded after 2 h of red-light illumination (625 nm, 100 mW/cm²) of the PdTPBP/PDI pair and the electron donor TEA in the degassed DMF solution (Fig. 4a). By contrast, only trace products could be detected in the control experiments that are either conducted with oxygen or without red-light irradiation, or in the absence of TEA or PDI or PdTPBP (Supplementary Table 5). These results confirmed that the energies of the two 625 nm photons are sufficient to complete the photoreduction of aryl bromides. This was a proof-of-concept of the feasibility for the long wavelength light driven photocatalysis using TS-conPET.

We then compared the photocatalytic efficiency of red-light mediated TS-conPET with conventional blue-light mediated conPET using the same model substrate. The product yield under 2-h blue-light

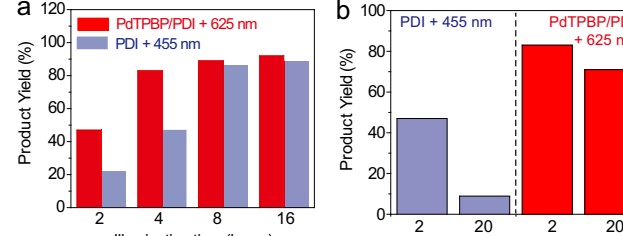
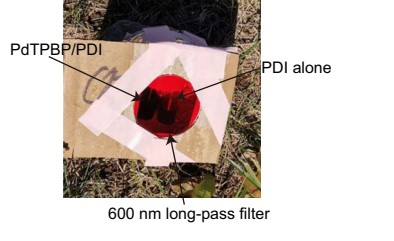

**Fig. 4 | Red light-driven TS-conPET process is superior than blue light-driven conPET process in view of the photoreduction rate and penetration depth.** **a** Comparison of the product yields of the red-light (625 nm) and blue light (455 nm) LED driven photoreduction of 4-bromoacetophenone under different irradiation time (2, 4, 8, 16 h). **b** The product yields of different reaction volumes including 2 mL and 20 mL under blue (455 nm) and red-light (625 nm) illumination for 4 h. **c** The picture of sunlight driven photoreduction of 4-bromoacetophenone with a 600 nm long-pass filter to reject the light of wavelength shorter than 600 nm to enter into the reaction setup. The corresponding photocatalyst component of PdTPBP/PDI or PDI alone are marked.

illumination (455 nm, 100 mW/cm²) is 22%, which is much lower than the red-light mediated reaction (47%). Moreover, compared to blue light-driven reaction, we found that the photoreduction rate increase is also much faster for red light-mediated reaction. In particular, in 4 h, the photoreaction with red-light is nearly complete and reaches a plateau yield of 83%, while the one with blue-light illumination has only a 47% yield (Fig. 4a). These experimental results clearly demonstrated that TS-conPET strategy is superior in photocatalytic performance than the traditional conPET pathway.

The advantage of red-light mediated TS-conPET over conventional blue-light triggered conPET was further demonstrated by the photoreduction performance in enlarged-volume reactions. The setup of enlarged-volume reaction is shown in Supplementary Fig. 17. When the reaction volume expanded to 20 mL from original 2 mL, after 4 h of light irradiation, the photoreduction yield of 4-bromoacetophenone declined sharply (from 47% to 9%) for blue-light (455 nm) activated conPET meanwhile the red-light (625 nm) reaction still maintained a high yield of 71% (originally 83%) (Fig. 4b). This result showed the deep penetration superiority of our long-wavelength light mediated conPET in the large colored reaction medium. In addition, our TS-conPET method can overcome the photobleaching and photodamage problem found in the conventional blue-light mediated conPET process. More specifically, in prolonged reactions, obvious decomposition of PDI can be detected in the conventional conPET method but this did not occur in our TS-conPET system (Supplementary Figs. 18 and 19). Collectively, our results showed that this TS-conPET process excel in regard to its elevated photocatalytic rate and enhanced reaction penetration depth, with much reduced photodamage when compared to the traditional conPET process.

More interestingly, due to the potent energy-utilization efficiency of the TS-conPET process, in addition to the 625 nm light, we found that 650 nm light, which is the absorption tail of PdTPBP, can also efficiently trigger the photoreduction of 4-bromoacetophenone. We envision that the utilization of 650 nm as excitation light would further avert possible photobleaching of PdTPBP, as its absorption is over ten times weaker ($\varepsilon = 8440\ M^{-1}\ cm^{-1}$) at 650 nm than that of 625 nm ($\varepsilon = 98000\ M^{-1}\ cm^{-1}$). In addition, the absorbance of PDI·⁻ at 650 nm is stronger than that of 625 nm (Supplementary Fig. 20), which should facilitate the second PET in the TS-conPET strategy. These hypotheses were confirmed by the following experimental results (Fig. 5a). After 1 h of irradiation by 625 nm light, the phosphorescence intensity of PdTPBP decreased to 60% of the original value, showing obvious photobleaching. By contrast, the phosphorescence intensity of PdTPBP remained constant when irradiated by 650 nm light, demonstrating that 650 nm light minimizes photodamage. The generation of PDI·⁻ under 650 nm illumination was then monitored with absorption spectra. After 1 min irradiation with 650 nm light, obvious peaks of PDI·⁻ can be detected for the PdTPBP/

PDI pair (Supplementary Fig. 21). Importantly, a high reduction yield of 4-bromoacetophenone, 79%, was obtained in a 2 mL reaction after 4 h of irradiation with 650 nm LED (100 mW/cm²) (Fig. 5b). This photoreduction yield is comparable to the 625 nm-irradiated reaction (83%). These results demonstrated that the increased photostability of PdTPBP and the stronger absorption of PDI·⁻ at 650 nm than that of 625 nm can effectively balance the effect of the inferior absorption of PdTPBP at the far-red light region. Thus, the photocatalytic mechanism of this efficient long-wavelength photoreduction of aryl halides was outlined in Fig. 5c. Firstly, the absorption of first red photon by PdTPBP gives the triplet excited PdTPBP (³PdTPBP*), which sensitize PDI to its triplet (³PDI*). Then the first PET occurs between ³PDI* and electron donor Et₃N to obtain PDI·⁻. The second red photon then excite PDI·⁻ to its excited state (²PDI·⁻*), which is capable of reducing inert aryl halides (Ar-X).

The excellent photocatalytic performance of our TS-conPET system under low-power long-wavelength LED light led us to investigate its applicability under ambient sunlight. As a proof-of-concept study, we conducted the photoreduction of 4-bromoacetophenone directly under outdoor sunlight, with a 600 nm long pass-filter placed in front of the reaction solution to exclude any interference from short-wavelength light (Fig. 4c). After 8 h of exposure to outdoor sunlight, we obtained a notable photoreduction yield of 27%, even without stirring the reaction solution. Notably, no detectable product was found in the control experiment conducted under the same conditions but without the photosensitizer PdTPBP. These results confirm the high energy-utilization efficiency of the TS-conPET system and its potential to operate effectively under ambient sunlight. This proof-of-concept study demonstrates the promising potential of the TS-conPET system for environmentally friendly and sustainable photocatalytic processes.

After successfully demonstrating the feasibility of photoreduction using long-wavelength light, we systematically investigated whether the TS-conPET method could be extended to other types of aryl halides. We selected substrates representing three typical groups of inert aryl halides (Fig. 5d). The first group consisted of simple aryl bromides with electron-withdrawing groups, including 4-bromobenzonitrile, 2-bromobenzonitrile, and 4-bromobenzaldehyde. This group exhibited excellent photoreduction yields (69–81%). The second group comprised pyrimidine and thiophene derivatives, which yielded good photoreduction yields of 76% and 56%, respectively. These results suggest that the TS-conPET process is efficient in the presence of heteroatom modifications. Finally, we tested a third group of aryl chlorides with the highest reduction potential, for which we achieved satisfying photoreduction yields of 49–57%. Therefore, these findings demonstrate that our long-wavelength activated TS-conPET approach is highly versatile and can be applied to a wide range of inert aryl halides for photoreduction. This has significant implications for

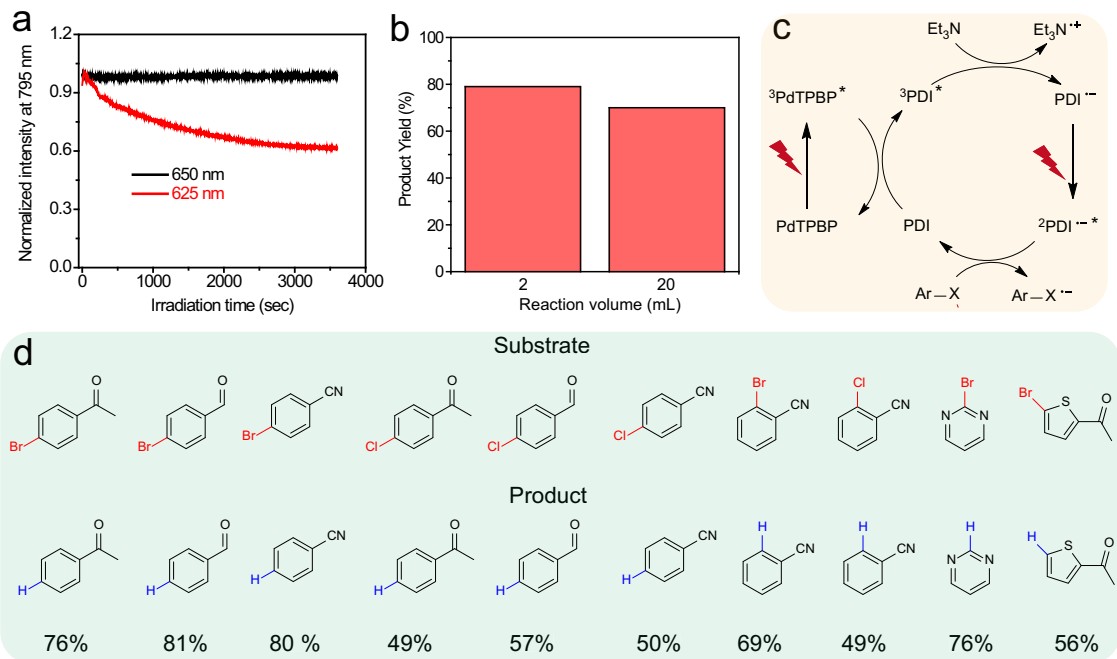

**Fig. 5 | Photoreduction of various aryl bromides and chlorides via TS-conPET process with far red light. a** Comparison of the luminescence intensity of PdTPBP under constant irradiation of 625 nm and 650 nm light. **b** The photoreduction yield of 4-bromoacetophenone at different reaction volumes including 2 mL and 20 mL when irradiated by far red light (650 nm) for 4 h. **c** Photocatalytic mechanism for the red light-driven reduction of aryl halides with PdTPBP/PDI pair. $^3$PdTPBP*, the excited triplet state of PdTPBP; $^3$PDI*, the excited triplet state of PDI; PDI$^{\cdot-}$, the radical anion of PDI$^{\cdot-}$; $^2$PDI$^{\cdot-}$*, the excited doublet state of PDI$^{\cdot-}$; Et$_3$N$^{+}$, the radical cation of Et$_3$N; Ar-X, aryl halide; Ar-X$^{\cdot-}$, the radical anion of aryl halide. **d** Far red light (650 nm)-driven TS-conPET process for the efficient photoreduction of various aryl halides. The red bonds highlight the reacted carbon halide bond, meanwhile, the blue bonds refer to the newly generated bond.

organic synthesis and underscores the potential of this method for various applications in the field.

## Expansion to the NIR light driven triplet sensitization-mediated conPET

The above-mentioned exciting results pushed us to explore the versatility of this TS-conPET strategy. Firstly, the guidelines for the construction of the TS-conPET PS/PC pair were summarized. As we have illustrated in Fig. 1b, the TS-conPET approach primarily consists of the photosensitization step and the two consecutive PET process. The key component is the PC, which should possess the ability to generate stable and light-absorbing radical anion, as this is the fundamental requirement for conPET. In addition, this radical anion of PC needs to be produced from the triplet state of the PC. For the PS component, its triplet state energy level (T$_1$) should be higher than that of PC to ensure the smooth operation of triplet sensitization. Besides, to demonstrate the advantage of TS-conPET than normal conPET, the singlet state energy level (S$_1$) of the PS should be lower than that of PC, enabling the utilization of low-energy (long-wavelength) photons. According to our current understanding, there is possibility to expand the application of TS-conPET by integrating other conPET photocatalysts. We recently discovered, for instance, that thioethyl-substituted naphthalene diimide (NDI) and its metal-organic frameworks (MOFs) exhibit conPET performance[41], and the TS-conPET system might be developed by pairing this NDI PC with photosensitizers such as PtOEP and PdTPBP. Furthermore, Wenger et al. recently reported that the reported conPET PC of 9,10-dicyanoanthracene can be sensitized to its triplet state by a red-light absorbing copper complex and then produce the radical anion[26].

Particularly, for the conPET photocatalyst PDI, we found that photosensitizers with higher T$_1$ than PDI, such as metal-free red light-absorbing BDP (T$_1$ = 1.43 eV)[42] and near-infrared (NIR) light-absorbing PtTNP (T$_1$ = 1.44 eV)[43] can replace the role of PdTPBP (Fig. 1c and

Supplementary Fig. 22, Supplementary Table 1). In the presence of TEA, obvious absorption signals of PDI radical anion were detected for the mixture of BDP/PDI upon 650 nm irradiation (Fig. 6a) and the mixture of PtTNP/PDI upon 721 nm irradiation (Fig. 6b). These results are promising and desirable for the operation of long-wavelength light (both red light and NIR light)-driven atom transfer radical polymerization (ATRP). ATRP has been proven to be one of the most efficient methods of controlled free radical polymerization[44] and the photoredox ATRP with low-energy photon owns the advantages of low-temperature operation, high response, optical control, and low polymer dispersity[45,46]. Very recently, Hadjichristidis N. et al. utilized the conPET property of PDI to achieve the ATRP of methyl methacrylate (MMA) with the initiator ethyl 2-bromopropionate (EBrP), which exhibited lower dispersity (1.22 vs 1.69) and higher initiator efficiency (66.7% vs 9.8%) than the one-photon pathway[46]. Based on this information, we explored the ATRP potential of our TS-conPET system with the gel formation reaction with MMA, EBrP, and the crosslinker ethylene glycol dimethacrylate (EGD).

Firstly, the metal-free pair BDP/PDI was devoted to the DMF solution of MMA:EGD:EBrP:TEA (molar ratio 20:2:1:1, see Methods section for details) with a final concentration of 17 mM for BDP and 0.5 mM for PDI, respectively. As shown in Supplementary Fig. 23, the generation of free-standing gel was detected after 1 h treatment of this system with far red light (650 nm, 40 mW cm$^{-2}$). Control experiments verified that photosensitizer BDP, photocatalyst PDI, and red excitation are all indispensable for this long-wavelength fabrication of gel (Supplementary Fig. 24). The gel-trigger wavelength can be further shifted to NIR (721 nm) with PtTNP as the photosensitizer. Notably, we found that the polymerization initiator can be upgraded to 4-bromoacetophenone, which own a reduction potential of nearly 1.0 V higher than that of EBrP. As shown in Fig. 6c, gel formation was detected after 4 h of NIR excitation (721 nm, 40 mW cm$^{-2}$), indicating the successful operation of ATRP by PtTNP/PDI with

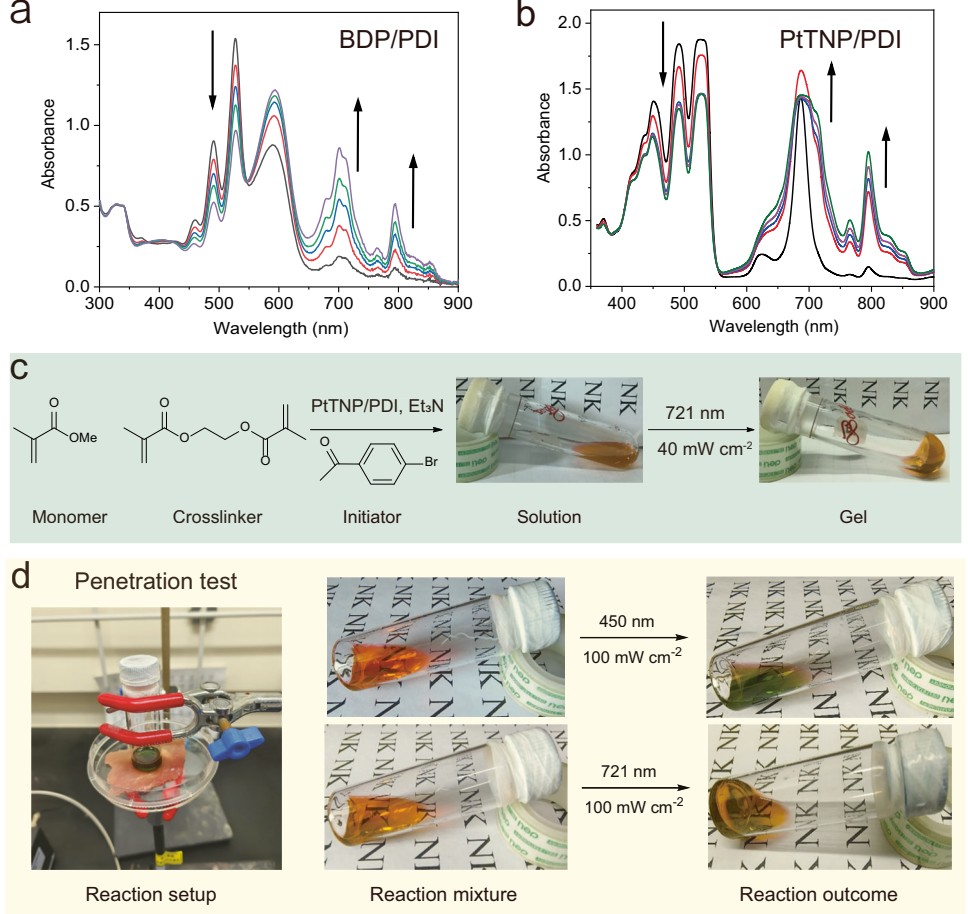

**Fig. 6 | Demonstration of the versatility of this TS-conPET strategy.** UV-vis absorption changes of (**a**) BDP/PDI in DMF solution with Et₃N upon red light irradiation or (**b**) PtTNP/PDI in DMF solution with Et₃N upon NIR light irradiation. Arrows indicated the absorption change direction along with the irradiation time. **c** NIR-driven photopolymerization to produce gel with PtTNP/PDI pair via TS-conPET process. **d** Penetration test of the photopolymerization with the barrier of a pork loin of 1.5 mm. Pictures of the setup and the reactions with blue (450 nm) or NIR (721 nm) light are displayed.

4-bromoacetophenone. Furthermore, the penetration superiority of this NIR ATRP system was verified by placing a pork loin of 1.5 mm in front of the light source (Fig. 6d). Using the same ATRP setup, the reaction mixture excited by blue light just showed the change of solution color, while the one with NIR excitation still complete the fabrication of gel. These results suggested that TS-conPET of BDP/PDI or PtTNP/PDI is efficient for photocatalytic ATRP with long-wavelength light and initiators of a broad range, which is of great research importance in 3D printing, laser direct writing, and micro- and nano-patterning.

In conclusion, our study presents an interesting and versatile TS-conPET approach that can accelerate the generation speed of stable radical anion by over 100 times compared to the state-of-the-art con-PET method. This is achieved by utilizing the triplet excited state of the photocatalyst PDI, which was previously unattainable using existing conPET methods. Additionally, the TS-conPET approach enables the excitation limit of conPET to be shifted to the NIR range, which enhances energy-utilization efficiency by reducing the energy loss of the first PET step and facilitating the second PET step. This highly efficient generation of stable radical anions enables long-wavelength photoreduction of aromatic C-Br or C-Cl bonds, significantly improving the photoreduction reaction rate of traditional conPET processes, especially for large-scale photocatalytic reactions. Additionally, the metal-free TS-conPET pair and the NIR light-absorbing TS-conPET pair were utilized for the atom transfer radical polymerization of MMA, which was highly desirable for material fabrication inside biological

systems. Our conceptual work opens up opportunities for the widespread application of long-wavelength-driven photoredox catalysis, which has been limited by current approaches.

## Methods

### Photoreduction of 4-bromoacetophenone with PDI

Photocatalyst PDI (2.0 mg, 1.25 mM), electron donor TEA (57 μL, 200 mM) and 4-bromoacetophenone (10 mg, 25 mM) were mixed in anhydrous DMF (2 mL) and then degassed for at least 15 min with argon. Then, the solution was excited by a deep blue LED (455 nm, 100 mW/cm²) at 40 °C. After the photoreduction, the raw product was diluted with ether (3 mL), and then 3 mL 1 mol/L HCl was added. This mixture was stirred for 3 min, and the upper organic layer was collected. 3 mL saturated sodium chloride solution was added to the organic layer to wash the HCl. The upper ether layer was collected, and then removed the ether. The product yields were measured via gas chromatography with internal standard of benzonitrile (Supplementary Figs. 29–33). The GC spectrum of the standard sample consisting of the equivalent amount of benzonitrile, acetophenone and 4-bromoacetophenone, is provided as Supplementary Fig. 35.

### Photoreduction of 4-bromoacetophenone with PdTPBP/ PDI pair

Photosensitizer PdTPBP (50 μM), photocatalyst PDI (2.0 mg, 1.25 mM), electron donor TEA (57 μL, 200 mM) and 4-bromoacetophenone (10 mg, 25 mM) were mixed in anhydrous DMF (2 mL) and then

degassed for at least 15 min with argon. Then the solution was excited with a red-light LED (625 nm) or far red-light LED (655 nm) (100 mW/cm$^2$) at 40 °C. After completed reaction, the raw product was diluted with ether (3 mL), and then 3 mL 1 mol/L HCl was added. This mixture was stirred for 3 min, and the upper organic layer was collected. 3 mL of saturated sodium chloride (NaCl) solution was added to the organic layer to wash the HCl. The upper ether layer was collected, and then removed the ether. The product yields were measured via gas chromatography with internal standard of benzonitrile (Supplementary Figs. 25–28 and 34).

### Photoreduction of other aryl halides with PdTPBP/PDI pair

Photosensitizer PdTPBP (50 μM), photocatalyst PDI (2.0 mg, 1.25 mM), electron donor TEA (57 μL, 200 mM), and aryl halides (10 mg, 25 mM) were mixed in anhydrous DMF (2 mL) and then degassed for at least 15 min with argon. Then the solution was excited with a far red-light LED (650 nm, 100 mW/cm$^2$) at 40 °C. After 16 h reaction, the raw product was diluted with ether (5 mL), and then 5 mL 1 mol/L HCl was added. This mixture was stirred for 3 min, and the upper organic layer was collected. Add 5 mL of saturated sodium chloride (NaCl) solution to the organic layer to wash the HCl. The upper ether layer was collected, and removed the ether. The raw product was purified with small silica column.

### Long-wavelength light-driven photopolymerization to produce gel via TS-conPET process

Firstly, the monomer of MMA and the crosslinker of ethylene glycol dimethacrylate were purified through a small column of aluminum oxide to remove the stabilizer. Then, photosensitizer PtTNP or BDP (0.034 μmol), photocatalyst PDI (0.7 mg, 1 μmol), electron donor TEA (69 μL, 0.5 mmol), MMA (1.07 mL, 10 mmol), EGD (188 μL, 1 mmol), EBrP (65 μL, 0.5 mmol) and anhydrous DMF (0.5 mL) were mixed in a tube within the glove box and then sealed to be kept in a deoxygenated atmosphere. Then the solution was excited with NIR light (for PtTNP, 721 nm, 20 mW/cm$^2$) or far-red light (for BDP, 650 nm, 20 mW/cm$^2$) at room temperature for 1 h.

## Data availability

All data pertaining to this manuscript are available within the main text and Supplementary Information. All relevant data underlying the results of this study are available from the corresponding authors upon request.

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

## Acknowledgements
Financial support from the National Natural Science Foundation of China (22371135 to L. Zeng, 21820102001 and 21890381 to C. Duan), and the Postdoctoral Fellowship Program of CPSF (GZB20230315 to L.Z.). G.H., K.H., and L.H. were exclusively supported by University of Massachusetts Chan Medical School. The authors thank Dr. Zhaolong Wang and Prof. Kaifeng Wu (Dalian Institute of Chemical Physics) for assistance with obtaining ns-TA spectra of 630 nm excitation. The authors would also like to thank Zhenghao Li and Prof. Fei Wang (Nankai University, College of Chemistry) for assistance with the GC experiments.

## Author contributions
L.Z. and L.H. conducted the majority of experiments and ran all the data analysis. L.H. prepared the supplementary information, L.Z. wrote the manuscript. Z.H. conducted photopolymerization experiments. T.M. ran the measurement of the transition absorption spectra. K.H. did the Transmission Electron Microscope measurement. C.D. supervised this project and provided financial support. G.H. conceived and supervised this project and also co-wrote the manuscript.

## Competing interests
The authors declare no competing interests.
