## [Peer Review File · Nature Communications]

Reviewers' comments:

Reviewer #1 (Remarks to the Author):

Han and co-workers developed a triplet sensitization-mediated consecutive photoinduced electron transfer (TS-conPET) mechanism to capture and relay long-wavelength far-red photon energy, which enables efficient photocatalytic processes that can harness a wider range of light wavelengths. Previous photocatalytic techniques suffered from short-wavelength absorption of photocatalysts, suboptimal energy utilization efficiency, incompatibility with large-scale reactions, and potential photodamage. The study used the PdTPBP/PDI pair as an example to complete the TS-conPET cycle, which showed over 100-fold enhanced PDI radical anion ($\text{PDI}^{\bullet-}$) generation rate under long-wavelength light excitation with their triplet sensitization method compared to the conventional conPET under blue light irradiation. TS-conPET has substantial implications for various fields, particularly for large-scale photocatalytic reactions, providing an environmentally friendly and sustainable photocatalytic process. Overall, this work presents a novel and appealing approach for photoredox catalysis that can overcome limitations of existing conPET methods and using long wavelength light for chemical transformations. I therefore support its publication in Nature Communications after minor revisions.

a) The TS-conPET system comprises two components: a photosensitizer (PS) and a photocatalyst (PC). Are there any principles or guidelines for choosing the appropriate PS/PC pair? Is it possible to extend the TS-conPET system to other conPET photocatalysts besides the PdTPBP/PDI pair?

b) Page 3. The claim that current conPET systems are limited by the inherent absorption of photocatalysts and cannot extend beyond 500 nm, let alone reach the far-red region (600-700 nm) is inaccurate. A report (10.1021/jacs.1c05994) related to conPET has demonstrated that aryl chlorides can be activated using 525 nm light (as shown in the SI section). This study need to be properly cited.

c) All the substrates in this study are electron-deficient and therefore expected to easily undergo hydrodehalogenation, some only yield about 50%. Could you provide an explanation for this, such as incomplete raw material reaction or any accompanying side reactions? How about electron-rich substrates?

d) To further demonstrate the excellent photocatalytic performance of TS-conPET system and strengthen the synthetic utility, I'd like to suggest the author to explore more types of reactions other than hydrodehalogenation reactions.

Reviewer #2 (Remarks to the Author):

Han, Duan and co-workers report in their current manuscript a triplet-sensitized mechanism for the generation and re-excitation of a perylene diimide (PDI) radical anion. Following a current trend, this approach allows them to achieve high photoredox reactivities using two low-power photons. They present several mechanistic studies including time-resolved spectroscopic techniques and mechanistic irradiation experiments to highlight advantages over the conventional conPET strategy with PDI. Finally, several activated aryl chlorides were successfully dehalogenated via the "novel" strategy. It is indeed an interesting paper and several parts of the exp. work are carried out in a convincing way. However, the authors have overlooked some key papers in the field as detailed below, and thus overestimated the importance of their findings significantly. Publication in Nature Communications is therefore not recommended and I suggest Communications Chemistry as a suitable publishing medium after correcting several additional mistakes and addressing some issues.

Major issues:

1) Introduction: "As a result, current conPET systems are restricted to the inherent absorption of these photocatalysts, and none can extend beyond 500 nm, let alone reach the far-red region (600-700 nm)."

This is incorrect. The very same mechanism claimed here as new was initially published in 2016 and further more recent papers use the very same strategy with green or even red light. Han and Duan seem to be familiar with the work of the pertinent authors (Goetz and

Wenger, see references below) but selectively cited work that used blue or violet light.

The introduction has to be modified significantly to explain the state-of-the-art correctly.

-Chem. Sci., 2016,7, 3862-3868

(TS-conPET is certainly not new)

- <https://doi.org/10.1002/anie.201711692>

(green-light driven without further sensitizer)

-JACS Au 2022, 2, 6, 1488–1503

(TS-conPET with red light)

These references should be added to Table S6 as well.

Again, I stumbled onto "By utilizing this innovative mechanism,..." (page 5). It can certainly not be written like that.

2) Clear evidence exists for the photoactivity of decomposition products of the very same catalyst (Phys. Chem. Chem. Phys., 2018,20, 8071-8076). Could it be the case that this photoactive decomposition product is also produced via the suggested triplet pathway?

3) "We envision that this result is likely due to the better electron communication between electron donor and the long-lived 3PDI*"

This is called cage escape, isn't it? And it is well established that

a photoinduced electron transfer with a triplet has an inherently higher quantum yield compared to the

corresponding singlet quenching event. Please have a look at the review written by Kavarnos and Turro in

the 1980s.

4) Table S2, Fig. 2 Fig. S2 ...:

Sensitizer-acceptor ratios are completely useless for kinetic studies. Absolute concentrations need to be presented (as well) in the whole study to avoid confusion.

5) Fig. 3 and Figs. S13 and S14:

Would it be possible to calculate reliable quantum yields for the formation of the PDI radical anion under direct blue-light and indirect red-light conditions?

Nevertheless, the comparison with the 102-fold enhancement seems to be solid as standardized conditions were used.

When I compare these results to the photocatalysis experiments in Fig. 4 it becomes obvious that the conversion enhancement with the red-light approach is rather moderate under application-related conditions.

This could be a hint for the activity of decomposition products rather than the PDI radical anion. Clearly, PDI radical anion generation is not the rate-determining step.

A critical referee and someone thinking about the actual usefulness of the results/the new system would say "Why should we add a precious Pd compound to a reaction that perfectly works with a purely organic system and low-power visible light to drive simple dehalogenations, which belong to the most boring reactions?". Sorry for stating this in that way, but it probably highlights why the study is not suitable for NatComm in my view, considering also that the concept is not as new as the authors claim.

6) Fig. 2b and corresponding text:

It is a pity that the data quality is that low. It is almost impossible to see the PDI triplet.

They wrote it in a careful way, what I appreciate.

"These experimental results suggested that PdTPBP/PDI can conduct an efficient TTET process under different conditions to

produce 3PDI*"

„and a relatively weak ESA band in the range of 550–600 nm“

This is rather noise than an ESA band. According to Castellano and coworkers a significant ESA between 450 and 500 should be visible

(citation 27)

Additional experiments under optimized conditions

(i.e. with red excitation instead of 355 nm) should be carried out to obtain high-quality data sets and to provide unambiguous evidence

for the suggested mechanism. Laser studies with a 3-component system (sensitizer+catalyst+donor) would be very convincing and I would expect them

for a paper that is submitted to a high-impact journal (e.g. as presented for the very same mechanism in Chem. Sci., 2016,7, 3862-3868).

7) page 9 and SI:

"be -0.23 eV, revealing that it is thermodynamically favorable to generate"

With the numbers shown in page 3 of the SI document, I think

it is uphill with the given potentials and energies (+0.23 eV) and not downhill (-0.23 eV).

Please have another look at it.

Minor issues or mistakes that can be corrected rapidly:

1) Fig. 1b: The excited radical anion is a doublet state and not a singlet state (1->2).

2) page 5: $M-1 \text{ cm}^{-1} \rightarrow M^{-1} \text{ cm}^{-1}$

3) According to the IUPAC Gold Book the equation used for the estimation of the driving force of PET should not be called the Rehm-Weller equation.

A Rehm-Weller analysis is e.g. shown in the Schanze paper that they cited (Fig. 3b in ref. 20).

4) Table S4: are these one-point Stern-Volmer analysis datasets?

What is the concentration of TEA and substrate for the respective catalysis?

DMF+TEA: same rate constant but different quenching efficiency? Even if this would be correct, this is not

understandable in the current form.

5) The product characterization/yield determination is not acceptable in the current form.

To give an example: "The methyl group of substrates, 4-bromoacetophenone located at 2.58 ppm meanwhile the methyl group of products, acetophenone located at 2.62 ppm. Via integrating the areas of

these two peaks. The product yield was calculated." I would like to see the corresponding spectra with a clear

separation of both peaks. (By the way, all analytical data should be presented)

Did the authors use an internal standard?

Reviewer #3 (Remarks to the Author):

The submitted manuscript describes triplet sensitization-mediated consecutive photoinduced electron transfer (TS-conPET), which enables long-wavelength light (625 and 650 nm) to drive photoreduction of aryl halides. The manuscript is well-written, and the results are convinced enough to support their point of view on the performance of the photocatalytic system PdTPBP/PDI. This reviewer supports its acceptance by Nature Communications after addressing the following concerns.

1. The authors noted that both the photocatalyst PDI and photosensitizer PdTPBP were reported previously: "Firstly, PDI is one of the most established photocatalysts in the conventional conPET study. Secondly, on the photosensitizer side, PdTPBP has intense absorbance in the long-wavelength region." Has this PdTPBP/PDI system been studied previously for photocatalytic systems? Has long-wavelength excitation been attempted for this system before?

2. In "Fig. 3c, PDI in Ar, 625 nm LED illumination (20 mW/cm²). c (PdTPBP) = 10 μM, c (PDI) = 50 μM", the inset legend is different from the caption. Also, the corresponding explanation is not very clear.

3. It is better to show the cyclic voltammogram of PDI and present the calculation details of the redox power of its excited states.
4. The resolution of Fig. 4c is not good enough.
5. It's recommended to draw a mechanistic cycle within Fig. 5 for photocatalytic dehalogenation of aryl halides.
6. A few direct two-photon absorbing photocatalytic systems have been reported recently which can operate under deep red and near infrared light irradiation. They authors may need to revise the introduction section a little bit to reflect the current status of this field.

Reviewers' comments:

Reviewer #1 (Remarks to the Author):

Han and co-workers developed a triplet sensitization-mediated consecutive photoinduced electron transfer (TS-conPET) mechanism to capture and relay long-wavelength far-red photon energy, which enables efficient photocatalytic processes that can harness a wider range of light wavelengths. Previous photocatalytic techniques suffered from short-wavelength absorption of photocatalysts, suboptimal energy utilization efficiency, incompatibility with large-scale reactions, and potential photodamage. The study used the PdTPBP/PDI pair as an example to complete the TS-conPET cycle, which showed over 100-fold enhanced PDI radical anion (PDI^{•-}) generation rate under long-wavelength light excitation with their triplet sensitization method compared to the conventional conPET under blue light irradiation. TS-conPET has substantial implications for various fields, particularly for large-scale photocatalytic reactions, providing an environmentally friendly and sustainable photocatalytic process. Overall, this work presents a novel and appealing approach for photoredox catalysis that can overcome limitations of existing conPET methods and using long wavelength light for chemical transformations. I therefore support its publication in Nature Communications after minor revisions.

Reply: We thank the reviewer for the positive comments on this manuscript.

The TS-conPET system comprises two components: a photosensitizer (PS) and a photocatalyst (PC). Are there any principles or guidelines for choosing the appropriate PS/PC pair? Is it possible to extend the TS-conPET system to other conPET photocatalysts besides the PdTPBP/PDI pair?

Reply: We thank the reviewer for this instructive comment. There are guidelines for the construction of the TS-conPET PS/PC pair. As we have illustrated in Fig. 1b, the TS-conPET approach primarily consists of the photosensitization step and the two consecutive PET process. The key component is the PC, which should possess the ability to generate stable and light-absorbing radical anion, as this is the fundamental requirement for conPET. In addition, this radical anion of PC needs to be produced from the triplet state of the PC. For the PS component, its triplet state energy level (T1) should be higher than that of PC to ensure the smooth operation of triplet sensitization. Besides, to demonstrate the advantage of TS-conPET than normal conPET, the singlet state energy level (S1) of the PS should be lower than that of PC, enabling the utilization of low-energy (long-wavelength) photons.

According to our current understanding, there is possibility to expand the application of TS-conPET by integrating other conPET photocatalysts. We recently discovered, for instance, that thioethyl-substituted naphthalene diimide (NDI) and its metal-organic frameworks (MOFs) exhibit conPET performance (*Nat. Commun.* **2023**, 14, 4002), and the TS-conPET system might be developed by pairing this NDI PC with photosensitizers such PtOEP and PdTPBP. Furthermore, Oliver et al. recently reported that the reported conPET PC of 9,10-dicyanoanthracene can be sensitized to its triplet state by a red-light absorbing copper complex and then produce the radical anion (*JACS Au* **2022**, 2, 1488–1503). In brief, the TS-conPET serves as a broader and more comprehensive approach for expanding the range of optical excitation wavelengths in photocatalysis. Thus, we believe that more TS-conPET systems can be explored in accordance with the above-mentioned principles of molecular design.

(1) Page 3. The claim that current conPET systems are limited by the inherent absorption of photocatalysts and cannot extend beyond 500 nm, let alone reach the far-red region (600-

700 nm) is inaccurate. A report (10.1021/jacs.1c05994) related to conPET has demonstrated that aryl chlorides can be activated using 525 nm light (as shown in the SI section). This study need to be properly cited.

Reply: We're grateful for the reviewer's reminder. In the revised manuscript, we have corrected the mentioned section and cited the relevant literature (Ref 18) in the main text as well as in Table S6.

- (2) All the substrates in this study are electron-deficient and therefore expected to easily undergo hydrodehalogenation, some only yield about 50%. Could you provide an explanation for this, such as incomplete raw material reaction or any accompanying side reactions? How about electron-rich substrates?

Reply: We thank the reviewer to give us the opportunity for the explanation. As we can see in Fig. 5, the substrates that give the yield about 50% are the aryl chlorides and 2-acetyl-5-bromothiophene, which own higher reduction potentials and stronger carbon halide bonds. Thus, the reduction proceeds slower than other substrates and incomplete raw materials are detected from the gas chromatography.

We tested electron-rich substituted bromobenzenes, such as 4-methoxybromobenzene and 4-*N,N*-dimethylaminobromobenzene, per the reviewer's suggestion. Unfortunately, we did not observe photoreductive dehalogenation product, as in consistent with the reported work (*Science* **2014**, 346, 725). The excited state of the PDI radical anion cannot reduce the electron-rich substituted aryl bromides, as previously described in the literature (*J. Am. Chem. Soc.* **2020**, 142, 2204), and no product formation was thus seen.

- (3) To further demonstrate the excellent photocatalytic performance of TS-conPET system and strengthen the synthetic utility, I'd like to suggest the author to explore more types of reactions other than hydrodehalogenation reactions.

Reply: We appreciate the referee's advice. In our paper, the reductive hydrodehalogenation is just a model reaction for us to illustrate the ability of our TS-conPET pair to achieving high reducing power. The efficient hydrodehalogenation of a series of aryl halides indicate the synthetic potential of our TS-conPET for the generation of aromatic radical and the subsequent C-C/C-P/C-S/C-B bond formation, as we shown in our recent work (*Nat. Commun.* **2023**, 14, 4002).

To further demonstrate the benefit and versatility of our TS-conPET approach, we tried the long-wavelength light (both red light and NIR light)-driven atom transfer radical polymerization (ATRP). ATRP has been proven one of the most efficient methods of controlled free radical polymerization (*Science* **2005**, 309, 1200) and the photoredox ATRP with low-energy photon own the advantages of low-temperature operation, high response, optical control and low polymer dispersity (*Chem. Rev.* **2022**, 122, 1830; *J. Am. Chem. Soc.* **2023**, 145, 12737). To demonstrate our TS-conPET approach for ATRP, we firstly expanded the original PS/PC pair of PdTPBP/PDI to metal-free pair BDP/PDI and NIR-responsive PtTNP/PDI pair (Fig. 1c). UV-vis absorption investigation confirmed the successful generation of PDI radical anion under red (NIR) light irradiation in the presence of BDP/PDI or PtTNP/PDI (Fig. 6a and 6b). For ATRP setup, we choose the photoinitiator ethyl 2-bromopropionate (EBrP) with a reduction potential lower than -0.8 V (vs SCE), monomer of methyl methacrylate (MMA), crosslinker of ethylene glycol dimethacrylate (EGD) in DMF with the assistance of TEA to show the polymerization visibly via gel formation. As shown in Fig. 6c and Fig. S20, after one hour irradiation of red light or NIR light, the formation of organogel was detected for ATRP with PtTNP/PDI or BDP/PDI. While for the control experiments operating without light, photoinitiator or PDI, no organogel was produced. Hadjichristidis N. et al has confirmed that the ATRP of MMA in

DMF with TEA and EBrP requires the two-photon excitation of PDI (*J. Am. Chem. Soc.* **2023**, *145*, 12737). These results suggested that TS-conPET of BDP/PDI or PtTNP/PDI is required for this photocatalytic ATRP, which is of great research importance in 3D printing, laser direct writing, and micro- and nano-patterning.

Reviewer #2 (Remarks to the Author):

Han, Duan and co-workers report in their current manuscript a triplet-sensitized mechanism for the generation and re-excitation of a perylene diimide (PDI) radical anion. Following a current trend, this approach allows them to achieve high photoredox reactivities using two low-power photons. They present several mechanistic studies including time-resolved spectroscopic techniques and mechanistic irradiation experiments to highlight advantages over the conventional conPET strategy with PDI. Finally, several activated aryl chlorides were successfully dehalogenated via the "novel" strategy. It is indeed an interesting paper and several parts of the exp. work are carried out in a convincing way. However, the authors have overlooked some key papers in the field as detailed below, and thus overestimated the importance of their findings significantly. Publication in *Nature Communications* is therefore not recommended and I suggest *Communications Chemistry* as a suitable publishing medium after correcting several additional mistakes and addressing some issues.

Reply: We sincerely appreciate the reviewer's insightful comments and the opportunity to address their concerns. We acknowledge that our initial manuscript did not adequately discuss the relevant works in the field, which may have led to confusion for the reviewer. In response to this, we have carefully revised the manuscript to include a comprehensive discussion and citation of the key literature, providing a clearer context for our work and highlighting the unique aspects of our approach, as well as providing new discoveries on the extension of our work to the NIR range.

While we recognize the importance of the previous studies, we respectfully emphasize that our work presents distinct advantages and significant advancements over the reported systems. Our TS-conPET strategy offers the first comprehensive design rules that can be applied to a diverse range of dye pairs and longer wavelengths. Remarkably, we have successfully demonstrated the versatility of this strategy by extending it to metal-free dye pairs and NIR-active PtTNP/PDI, a wavelength range that has not been reported by any previous ConPET work. This groundbreaking achievement challenges the long-held belief that ConPET is forbidden in the near-infrared range, paving the way for a new era of photoredox catalysis.

Furthermore, we have showcased the practical utility of our TS-conPET system by implementing it in the photocatalytic atom transfer radical polymerization (ATRP) to produce organogels. Most notably, our work achieved NIR-driven atom transfer radical polymerization using an inert aromatic halide as the initiator, highlighting the exceptional energy-utilization efficiency of the TS-conPET process. This breakthrough opens up exciting possibilities for advanced applications, such as 3D printing, tissue engineering, and polymerization within living cells.

To further strengthen our manuscript, we have included additional experimental results and provided point-by-point responses to the reviewer's comments. These efforts aim to demonstrate the robustness and generalizability of our TS-conPET strategy, addressing any concerns raised by the reviewer and providing a more comprehensive understanding of our work.

In light of these improvements and the potential for our work to advance the field, we hope that the reviewer can concur with our perspective on the significance and novelty of our work. We greatly appreciate the reviewer's time and consideration, and we look forward to the opportunity to further discuss our work and address any remaining concerns.

Major issues:

- (1) Introduction: "As a result, current conPET systems are restricted to the inherent absorption of these photocatalysts, and none can extend beyond 500 nm, let alone reach the far-red region (600-700 nm)."

This is incorrect. The very same mechanism claimed here as new was initially published in 2016 and further more recent papers use the very same strategy with green or even red light. Han and Duan seem to be familiar with the work of the pertinent authors (Goetz and Wenger, see references below) but selectively cited work that used blue or violet light. The introduction has to be modified significantly to explain the state-of-the-art correctly.

-Chem. Sci., 2016,7, 3862-3868 (TS-conPET is certainly not new) - <https://doi.org/10.1002/anie.201711692> (green-light driven without further sensitizer)

-JACS Au 2022, 2, 6, 1488–1503 (TS-conPET with red light) These references should be added to Table S6 as well.

Again, I stumbled onto "By utilizing this innovative mechanism,.." (page 5). It can certainly not be written like that.

Reply: Thank you for bringing this issue to our attention. We appreciate the opportunity to address your concerns and have revised the mentioned parts of our manuscript accordingly. We recognize the relevance of the three papers you identified regarding the TS-conPET process and apologize for not adequately acknowledging their contributions in our original manuscript. However, we respectfully maintain that our work represents an important step forward in the general application of this powerful approach.

The first two works (Chem. Sci., 2016, 7, 3862-3868 and Angew. Chem. Int. Ed., 2018, 57, 1078) demonstrate the visible light (green) TS-conPET using $[\text{Ru}(\text{bpy})_3]^{2+}$ as a photosensitizer or ConPET photocatalyst but are limited to aqueous systems and specific setups, which hinders their general applicability in organic media commonly used in photoredox catalysis. The third work (JACS Au., 2022, 2, 1488) reported a Cu complex ($[\text{Cu}(\text{dap})_2]^+$) and 9,10-dicyanoanthracene (DCA) system, but the main mechanism is PET between $[\text{Cu}(\text{dap})_2]^+$ and DCA, with triplet sensitization being a competitive pathway.

In contrast, our work stands out by providing the first general design rules for the TS-conPET strategy, applicable to a wide range of dye pairs and longer wavelengths. We have successfully expanded this strategy to metal-free dye pairs and NIR-active PtTNP/PDI (in our revised manuscript) demonstrating their application in photocatalytic ATRP for gel formation. Notably, we achieved NIR-driven atom transfer radical polymerization using inert aromatic halide as the initiator, showcasing the high energy-utilization efficiency of the TS-conPET process and its potential for advanced applications such as 3D printing, tissue engineering, and polymerization inside living cells.

Thus, our work establishes clear guidelines for the choice of photosensitizer and conPET photocatalyst in the TS-conPET system, extending the original PdTPBP/PDI pair to metal-free BDP/PDI and NIR-active PtTNP/PDI. These advancements demonstrate the

generality and practicality of our approach in organic media and showcase its potential for a wide range of applications.

We have revised our paper according to the comments provided by you, changed the title to "Long-wavelength Near-infrared and Red Light-driven Consecutive Photo-induced Electron Transfer for Highly Effective Photoredox Catalysis", and updated Table S6 to include the relevant references. We believe our work significantly advances the TS-conPET concept, opening up novel applications in chemistry, material science, and biology.

We sincerely hope that these revisions address your concerns and meet your expectations. We believe that our work makes a valuable contribution to the field by providing a general and practical approach to TS-conPET systems, and we look forward to your positive reconsideration of our manuscript.

(2) Clear evidence exists for the photoactivity of decomposition products of the very same catalyst (Phys. Chem. Chem. Phys., 2018,20, 8071-8076). Could it be the case that this photoactive decomposition product is also produced via the suggested triplet pathway?

Reply: We thank the Reviewer for give a hint about the photoactivity of decomposed PDI. Indeed, we have observed the decomposition of PDI from the color of the reaction mixture after long time irradiation of blue light (Fig. S12 in previous edition), and we emphasized this decomposition was not observed for red light photocatalysis, which displayed the advantage of our TS-conPET approach. Inspired by the mentioned ref (Phys. Chem. Chem. Phys., 2018, 20, 8071), we conducted the UV-vis absorption investigation to further determine whether TS-conPET causes photodegradation of PDI. Figure S16 demonstrates that when PdTPBP/PDI was exposed to red light in the presence of TEA, it was entirely converted to PDI radical anion. In contrast, upon blue light irradiation, the absorption of the mixture of TEA and PDI undergoes the intensity decrease and the diminish of the fine-structure, suggesting the decomposition of PDI.

(3) "We envision that this result is likely due to the better electron communication between electron donor and the long-lived $^3\text{PDI}^*$ "

This is called cage escape, isn't it? And it is well established that a photoinduced electron transfer with a triplet has an inherently higher quantum yield compared to the corresponding singlet quenching event. Please have a look at the review written by Kavarnos and Turro in the 1980s.

Reply: We thank the Reviewer for bring about this discussion while from our opinion, electron transfer between the triplet state and the substrate does not necessarily have a higher quantum efficiency. Except for the lifetime of the excited state, there are other factors influencing the efficiency of photoinduced electron transfer such as the thermodynamic driving force and reorganization energy (Marcus theory, related refs: *Nat. Chem.* 2016, 8, 603; *Adv. Mater.* 2015, 27, 2496; *J. Am. Chem. Soc.* 2014, 136, 15869). For a given photocatalyst, the T_1 state has a lower energy level than the S_1 state, meaning that the driving force for electron transfer in the T_1 state is reduced than that of the S_1 state. Thus, in the mentioned ref (*Chem. Rev.* 1986, 86, 401) by Kavarnos and Turro, they comment that 'the singlet states of many sensitizers are more effective in electron-transfer quenching than the respective triplet states'. Our discovery that the T_1 state of PDI can accept the electrons of TEA to form PDI radical anions ensures the occurrence of TS-conPET and unexpectedly show a higher efficiency than the PET involving the S_1 state.

(4) Table S2, Fig. 2 Fig. S2 ...:

Sensitizer-acceptor ratios are completely useless for kinetic studies. Absolute concentrations

need to be presented (as well) in the whole study to avoid confusion.

Reply: We thank the Reviewer for the kind reminder and we have labeled the absolute concentrations on the revised manuscript.

(5) Fig. 3 and Figs. S13 and S14: Would it be possible to calculate reliable quantum yields for the formation of the PDI radical anion under direct blue-light and indirect red-light conditions? Nevertheless, the comparison with the 102-fold enhancement seems to be solid as standardized conditions were used.

Reply: According to the Reviewer's suggestion, we determined the photogenerated PDI radical anion efficiencies of 0.5% and 48.3% for direct blue light irradiation and TS-conPET under red light irradiation, respectively, using previously reported methods (*J. Am. Chem. Soc.* **2014**, *136*, 9256). Please refer to Supporting Material for additional details.

(6) When I compare these results to the photocatalysis experiments in Fig. 4 it becomes obvious that the conversion enhancement with the red-light approach is rather moderate under application-related conditions. This could be a hint for the activity of decomposition products rather than the PDI radical anion. Clearly, PDI radical anion generation is not the rate-determining step. A critical referee and someone thinking about the actual usefulness of the results/the new system would say "Why should we add a precious Pd compound to a reaction that perfectly works with a purely organic system and low-power visible light to drive simple dehalogenations, which belong to the most boring reactions?". Sorry for stating this in that way, but it probably highlights why the study is not suitable for NatComm in my view, considering also that the concept is not as new as the authors claim.

Reply: We thank the Reviewer for giving us the opportunity to justify our results. As we respond to the second comment, for our TS-conPET strategy, there are no decomposition of PDI, which is an advantage over the traditional conPET. In addition, the photocatalytic comparison between red-light or blue-light reduction clearly shows that the red light system of TS-conPET give higher yield than the blue light system, especially in short-reaction period and large volume reaction (Fig. 4a and 4b). For the application-related conditions, which we think is the large-volume setup of 20 mL, the red-light system is 6-fold higher in reduction yield than the blue one. This result verified that fewer energy input can lead to higher production yield under specific conditions, which we believe is exciting.

Alternatively, we choose the dehalogenation as a model reaction to demonstrate the reducing power of the excited PDI radical anion. To provide much more appealing prospects of this TS-conPET strategy, in the revised manuscript, we devised the metal-free BDP/PDI system and the NIR-active PtTNP/PDI system (Fig. 1 and Fig. 6). More importantly, these new pairs of TS-conPET can effectively trigger the atom transfer radical polymerization (ATRP) to generate organogels (Fig. 6 and Fig. S20), which are applicable for 3D printing, laser direct writing, and biomedicine. These results not only demonstrate that the TS-conPET mechanism is highly generalizable, but also reveals its potential for functional materials fabrication. Thus, we believe that TS-conPET with red and even NIR

light excitation using PDI as a photocatalyst, as proposed in this manuscript, has significant research and application value.

(7) Fig. 2b and corresponding text:

It is a pity that the data quality is that low. It is almost impossible to see the PDI triplet.

They wrote it in a careful way, what I appreciate.

"These experimental results suggested that PdTPBP/PDI can conduct an efficient TTET process under different conditions to produce $^3\text{PDI}^*$ "

„and a relatively weak ESA band in the range of 550–600 nm“

This is rather noise than an ESA band. According to Castellano and coworkers a significant ESA between 450 and 500 should be visible (citation 27)

Additional experiments under optimized conditions (i.e. with red excitation instead of 355 nm) should be carried out to obtain high-quality data sets and to provide unambiguous evidence for the suggested mechanism. Laser studies with a 3-component system (sensitizer+catalyst+donor) would be very convincing and I would expect them for a paper that is submitted to a high-impact journal (e.g. as presented for the very same mechanism in Chem. Sci., 2016,7, 3862-3868).

Reply: Per the Reviewer's suggestion, we have conducted the transition spectra of PdTPBP/PDI with the excitation of 630 nm. As shown in Fig. 2, the triplet peaks of PDI in the range of 450~500 nm becomes obvious along with the lasting of the decay time. More importantly, when the donor molecule Et_3N was added, the newly-emerged GSB peak at 700 nm was detected (Fig. S10), suggesting the generation of PDI radical anion. These results provide solid evidence for the proposed TS-conPET mechanism.

(8) page 9 and SI: "be -0.23 eV, revealing that it is thermodynamically favorable to generate" With the numbers shown in page 3 of the SI document, I think it is uphill with the given potentials and energies (+0.23 eV) and not downhill (-0.23 eV). Please have another look at it.

Reply: We appreciate the reviewer's detailed and careful examination of this manuscript and pointing out this error. To more precisely calculate the electron transfer Gibbs free energy (ΔG_{ET}), we first determined the cyclic voltammetry profile of PDI to obtain its redox potential (Fig. S9). The T1 energy level of PDI (1.21 eV, *J. Am. Chem. Soc.* **2010**, *132*, 14203) was utilized in the calculation of ΔG_{ET} between TEA and $^3\text{PDI}^*$ to generate the PDI radical anion. The new ΔG_{ET} result is 0.24 eV. Meanwhile, our transient absorption spectrum (Fig. S10) and UV-vis absorption spectra analysis (Fig. 3) revealed the formation of PDI radical anion for the mixture of PdTPBP and PDI in the presence of TEA, indicating that electron transfer between the PDI triplet state and TEA is permitted in the setup.

Minor issues or mistakes that can be corrected rapidly:

1) Fig. 1b: The excited radical anion is a doublet state and not a singlet state (1->2).

Reply: We've corrected this error in the revised manuscript.

2) page 5: M-1 cm-1 -> M⁻¹ cm⁻¹

Reply: This typo has been corrected.

3) According to the IUPAC Gold Book the equation used for the estimation of the driving force of PET should not be called the Rehm-Weller equation. A Rehm-Weller analysis is e.g. shown in the Schanze paper that they cited (Fig. 3b in ref. 20).

Reply: We thank the Reviewer for pointing out this issue and we've updated the related sections.

4) Table S4: are these one-point Stern-Volmer analysis datasets?

What is the concentration of TEA and substrate for the respective catalysis?

DMF+TEA: same rate constant but different quenching efficiency? Even if this would be correct, this is not understandable in the current form.

Reply: We determined the Stern-Volmer quenching constants by measuring the luminescence quenching of PdTPBP at six different PDI concentrations (the source data are shown in Figs. S4-S5 and 2d) and fitting the k_{sv} constants based on Equation 1 and 2. So these results are not from a single-site analysis.

The concentration of TEA and the substrate (4-bromoacetophenone) during the k_{sv} test were all 1 mM (100 eq. of PdTPBP). For catalysis, as we shown in the Methods section, the concentration of PdTPBP is 50 μ M, while the concentration of TEA and 4-bromoacetophenone is 200 mM and 25 mM. We have updated these details in the revised manuscript.

Besides, the irrational data for the groups in Table S4 have been checked and recalculated.

5) The product characterization/yield determination is not acceptable in the current form.

To give an example: "The methyl group of substrates, 4-bromoacetophenone located at 2.58 ppm meanwhile the methyl group of products, acetophenone located at 2.62 ppm. Via integrating the areas of these two peaks. The product yield was calculated." I would like to see the corresponding spectra with a clear separation of both peaks. (By the way, all analytical data should be presented)

Did the authors use an internal standard?

Reply: The ¹H-NMR peaks of 4-bromoacetophenone and acetophenone could be separated from each other as shown in the following example.

However, to be more accurate, gas chromatography data with internal standard were provided to determine the yield (See Figs. S22-S31).

Reviewer #3 (Remarks to the Author):

The submitted manuscript describes triplet sensitization-mediated consecutive photoinduced electron transfer (TS-conPET), which enables long-wavelength light (625 and 650 nm) to drive photoreduction of aryl halides. The manuscript is well-written, and the results are convinced enough to support their point of view on the performance of the photocatalytic system PdTPBP/PDI. This reviewer supports its acceptance by Nature Communications after addressing the following concerns.

Reply: We are appreciative of the referee's evaluation for this research paper and will respond to the following comments point by point.

1. The authors noted that both the photocatalyst PDI and photosensitizer PdTPBP were reported previously: “Firstly, PDI is one of the most established photocatalysts in the conventional conPET study. Secondly, on the photosensitizer side, PdTPBP has intense absorbance in the long-wavelength region.” Has this PdTPBP/PDI system been studied previously for photocatalytic systems? Has long-wavelength excitation been attempted for this system before?

Reply: We thank the Reviewer for this enlightening comment. As far as we known, there are no report for the PdTPBP/PDI system for photocatalytic investigation or for long-wavelength excitation. There are no results for the searching of the combination between PdTPBP and PDI in either Web of Science or SciFinder. In contrast, there are lots of results for the combination between PdTPBP and perylene. This might due to the rare study of the triplet state of PDI.

2. In “Fig. 3c, PDI in Ar, 625 nm LED illumination (20 mW/cm²). c (PdTPBP) = 10 μM, c (PDI) = 50 μM”, the inset legend is different from the caption. Also, the corresponding explanation is not very clear.

Reply: We thank the reviewers for carefully reviewing the manuscript and we have made

revisions to the related caption.

3. It is better to show the cyclic voltammogram of PDI and present the calculation details of the redox power of its excited states.

Reply: Per the Reviewer's suggestion, we determined the CV of PDI and added the calculation details for the excited state redox potential of PDI in the revised supporting information. As shown in Fig. S9, $E_{\text{red}}(\text{PDI}/\text{PDI}^{\cdot-}) = -0.89 \text{ V (vs Fc/Fc}^+) = -0.44 \text{ V (vs SCE)}$, which is close to the reported value of $-0.43 \text{ V (vs SCE, } J. Am. Chem. Soc. \text{ 2020, } 142, 2204)$. The corresponding $E_{0,0}$ value of PDI triplet is 1.21 eV according to the ref (*J. Am. Chem. Soc.* **2010**, *132*, 14203). Thus, the redox power of the triplet excited state of PDI is calculated to be $+0.76 \text{ V (vs SCE)}$ using equation 5. These details have been added to the SI.

4. The resolution of Fig. 4c is not good enough.

Reply: The photo of Fig. 4c is a cell phone shot of the experimental set up for the sunlight-driven photoreduction. The sunlight exposure plus the 600-nm long-pass filter leads to the relative low resolution of the reactors inside.

5. It's recommended to draw a mechanistic cycle within Fig. 5 for photocatalytic dehalogenation of aryl halides.

Reply: We thank the referee for his/her suggestion and we have drawn a mechanistic diagram of photoreductive dehalogenation in Fig. 5.

6. A few direct two-photon absorbing photocatalytic systems have been reported recently which can operate under deep red and near infrared light irradiation. They authors may need to revise the introduction section a little bit to reflect the current status of this field.

Reply: We thank the Reviewer for this instructive suggestion and have rewritten the introduction section of this manuscript to be more consistent with current advances in the field.

REVIEWER COMMENTS

Reviewer #1 (Remarks to the Author):

Han, Duan, and co-workers reported a triplet sensitization-enabled consecutive photoinduced electron transfer (TS-conPET) process employing long-wavelength near-infrared (NIR) and red light to drive the dehalogenation of aryl halides and the atom transfer radical polymerization reaction. Notably, the PdTPBP/PDI pair exhibited over a 100-fold enhancement in PDI radical anion generation under long-wavelength light excitation with their triplet sensitization method, compared to conventional conPET under blue light irradiation. Overall, this work presents an appealing method that utilizes long-wavelength light for chemical transformations, effectively overcoming the limitations of existing conPET methods. The authors have thoroughly addressed the reviewers' questions. Thus, I support its publication in Nature Communications after minor revisions.

1. The authors employed the PdTPBP/PDI pair to catalyze the dehalogenation of aryl halides using red light. How does the PtOEP/PDI pair perform under NIR light?
2. The authors proposed that π - π interactions between PdTPBP and PDI facilitate intermolecular triplet energy transfer, as indicated by the formation of spherical nanoparticles observed by TEM. More evidence should be added to support this point.

Reviewer #2 (Remarks to the Author):

Han, Duan and co-workers now present a careful revision of the manuscript that I reviewed very thoroughly.

Not only did the authors correct essentially all mistakes that I discovered but they also present novel findings with additional sensitized conPET pairs (NIR-driven or purely organic) as well as novel applications

highlighting the advantages when low-energy photons are used. The quality of the spectroscopic results has been

improved as well. Now I am happy to support the publication of this impactful manuscript in NCOMMS.

Following an additional request by the editorial office, I had a look at the new sub-figure (Figure 5c) and it

is correct in my view as it nicely describes the underlying mechanism.

Minor mistakes in the revised version:

-Oliver et al- -> Wenger et al.

-reply to referee #2 (major issue #3):

In the case when both singlet and triplet photoredox reactions are thermodynamically feasible, then the

triplet pathway has a decisive advantage (as also observed in your manuscript). Please see Figure 9 and related

text in in the excellent review written by Jerry Meyer and Ludovic Troian-Gautier very recently

(<https://pubs.acs.org/doi/full/10.1021/acs.chemrev.3c00930>).

Hence, there is a clear advantage when a triplet is reduced and the sentence "which might originate from the better electron communication"

is incorrect in my view.

Reviewer #1 (Remarks to the Author):

Han, Duan, and co-workers reported a triplet sensitization-enabled consecutive photoinduced electron transfer (TS-conPET) process employing long-wavelength near-infrared (NIR) and red light to drive the dehalogenation of aryl halides and the atom transfer radical polymerization reaction. Notably, the PdTPBP/PDI pair exhibited over a 100-fold enhancement in PDI radical anion generation under long-wavelength light excitation with their triplet sensitization method, compared to conventional conPET under blue light irradiation. Overall, this work presents an appealing method that utilizes long-wavelength light for chemical transformations, effectively overcoming the limitations of existing conPET methods. The authors have thoroughly addressed the reviewers' questions. Thus, I support its publication in Nature Communications after minor revisions.

Reply: Thank you for acknowledging our efforts to refine this work. We will address your remaining questions one by one.

1. The authors employed the PdTPBP/PDI pair to catalyze the dehalogenation of aryl halides using red light. How does the PtOEP/PDI pair perform under NIR light?

Reply: This specific pair of PtOEP/PDI could not catalyze the dehalogenation of aryl halides under NIR light, since the photosensitizer PtOEP is not capable of absorbing NIR (Chemical Physics 2006, 330, 118).

2. The authors proposed that π - π interactions between PdTPBP and PDI facilitate intermolecular triplet energy transfer, as indicated by the formation of spherical nanoparticles observed by TEM. More evidence should be added to support this point.

Reply: Thank you for the instructive comment. In order to get more evidence that the π - π interaction promoted triplet energy transfer, we designed the following control experiments. We employed leucine-modified PDI (Leu-PDI) as a photocatalyst (**Figure R1** or **Figure S9**). For nanoparticle preparation, the concentration of photocatalyst is much higher than that of the photosensitizer. Accordingly, we speculate that the charge-charge repulsion of the carboxylic acid would suppress the molecular stacking between the Leu-PDI, which prohibit the formation of nanoparticle and then the interaction between Leu-PDI and PdTPBP. As expected, we did not see PdTPBP/Leu-PDI forming spherical nanoparticles by TEM (**Figure R2** or **Figure S10**). Next, we determined the triplet energy transfer efficiencies (Φ_{TET}) of PdTPBP/PDI and PdTPBP/Leu-PDI at the same concentration of the photosensitizer and photocatalyst. The Φ_{TET} of PdTPBP/PDI (84.3%) (**Table S4**) is higher than that of PdTPBP/Leu-PDI (53.7%) (**Figure R3** or **Figure S11**), which further supports that π - π stacking between PDIs or between PdTPBP and PDI is promising for the enhancement of triplet energy transfer.

Figure R1. Molecular structures of Leu-PDI and PDI.

Figure R2. TEM image of the PdTPBP/Leu-PDI pair. Dark region is the sodium phosphotungstic acid staining.

Figure R3. Phosphorescence emission spectra of PdTPBP (10 μM) without Leu-PDI and in the presence of Leu-PDI (8.0 μM), $\lambda_{\text{ex}} = 630 \text{ nm}$, in DMF.

Reviewer #2 (Remarks to the Author):

Han, Duan and co-workers now present a careful revision of the manuscript that I reviewed very thoroughly.

Not only did the authors correct essentially all mistakes that I discovered but they also present novel findings with additional sensitized conPET pairs (NIR-driven or purely organic) as well as novel applications highlighting the advantages when low-energy photons are used. The quality of the spectroscopic results has been improved as well. Now I am happy to support the publication of this impactful manuscript in NCOMMS.

Following an additional request by the editorial office, I had a look at the new sub-figure (Figure 5c) and it is correct in my view as it nicely describes the underlying mechanism.

Reply: We appreciate your feedback on our previous mistakes and your praise for our recent efforts to enhance the novelty and quality of this work. We will address the minor errors individually

Minor mistakes in the revised version:

-Oliver et al- -> Wenger et al.

Reply: Thank you for the remainder. We have changed this mistake.

-reply to referee #2 (major issue #3):

In the case when both singlet and triplet photoredox reactions are thermodynamically feasible, then the triplet pathway has a decisive advantage (as also observed in your manuscript). Please see Figure 9 and related text in the excellent review written by Jerry

Meyer and Ludovic Troian-Gautier very recently (<https://pubs.acs.org/doi/full/10.1021/acs.chemrev.3c00930>).

Hence, there is a clear advantage when a triplet is reduced and the sentence "which might originate from the better electron communication" is incorrect in my view.

Reply: Thank you for the insightful discussion. We gained valuable insights from the mentioned paper on photochemical cage escape. In the revised text, we have updated the sentence to include this critical process. Additionally, we have added this important reference.

REVIEWERS' COMMENTS

Reviewer #1 (Remarks to the Author):

The authors have addressed my comments and concerns. I support its publication in Nature Communications.